# Immersive Teleoperation via Collaborative Device-Agnostic Interfaces for Smart Haptics: A Study on Operational Efficiency and Cognitive Overflow for Industrial Assistive Applications

**DOI:** 10.3390/s25133993

**Published:** 2025-06-26

**Authors:** Fernando Hernandez-Gobertti, Ivan D. Kudyk, Raul Lozano, Giang T. Nguyen, David Gomez-Barquero

**Affiliations:** 1iTEAM Research Institute, Universitat Politècnica de València (UPV), 46022 Valencia, Spain; raulote@iteam.upv.es (R.L.); dagobar@iteam.upv.es (D.G.-B.); 2Wandelbots GmbH, Technische Univeristät Dresden, 01159 Dresden, Germany; 3Haptic Communication Systems, Technische Univeristät Dresden, 01062 Dresden, Germany; giang.nguyen@tu-dresden.de

**Keywords:** haptic teleoperation, immersive interfaces, device-agnostic platform, real-time feedback

## Abstract

This study presents a novel investigation into immersive teleoperation systems using collaborative, device-agnostic interfaces for advancing smart haptics in industrial assistive applications. The research focuses on evaluating the quality of experience (QoE) of users interacting with a teleoperation system comprising a local robotic arm, a robot gripper, and heterogeneous remote tracking and haptic feedback devices. By employing a modular device-agnostic framework, the system supports flexible configurations, including one-user-one-equipment (1U-1E), one-user-multiple-equipment (1U-ME), and multiple-users-multiple-equipment (MU-ME) scenarios. The experimental set-up involves participants manipulating predefined objects and placing them into designated baskets by following specified 3D trajectories. Performance is measured using objective QoE metrics, including temporal efficiency (time required to complete the task) and spatial accuracy (trajectory similarity to the predefined path). In addition, subjective QoE metrics are assessed through detailed surveys, capturing user perceptions of presence, engagement, control, sensory integration, and cognitive load. To ensure flexibility and scalability, the system integrates various haptic configurations, including (1) a Touch kinaesthetic device for precision tracking and grounded haptic feedback, (2) a DualSense tactile joystick as both a tracker and mobile haptic device, (3) a bHaptics DK2 vibrotactile glove with a camera tracker, and (4) a SenseGlove Nova force-feedback glove with VIVE trackers. The modular approach enables comparative analysis of how different device configurations influence user performance and experience. The results indicate that the objective QoE metrics varied significantly across device configurations, with the Touch and SenseGlove Nova set-ups providing the highest trajectory similarity and temporal efficiency. Subjective assessments revealed a strong correlation between presence and sensory integration, with users reporting higher engagement and control in scenarios utilizing force feedback mechanisms. Cognitive load varied across the set-ups, with more complex configurations (e.g., 1U-ME) requiring longer adaptation periods. This study contributes to the field by demonstrating the feasibility of a device-agnostic teleoperation framework for immersive industrial applications. It underscores the critical interplay between objective task performance and subjective user experience, providing actionable insights into the design of next-generation teleoperation systems.

## 1. Introduction

Immersive teleoperation systems have emerged as transformative tools across various domains, enabling users to remotely perform complex tasks with a high degree of precision and control. These systems [1,2] leverage advanced robotics, haptic feedback, and tracking technologies to replicate real-world interactions in virtual or remote environments, enhancing user engagement and operational efficiency. In industrial assistive applications [3,4], where accuracy and reliability are paramount, the design of teleoperation frameworks must account for not only technical performance but also holistic quality of experience (QoE). Traditional systems [5,6] often lack adaptability and are constrained by device-specific dependencies, limiting their versatility in dynamic and collaborative environments. This study addresses these challenges by introducing a collaborative, device-agnostic teleoperation system guided by the framework shown in Figure 1 and designed to optimize the QoE through both objective and subjective assessments. By focusing on modularity, interoperability, and user-centric metrics, this work aims to set a new benchmark for immersive teleoperation in industrial contexts.

### 1.1. Background and Motivation

The rapid advancement of immersive technologies and haptic interfaces has revolutionized teleoperation systems, enabling users to interact with remote environments as though they are physically present. This evolution has significant implications for industrial assistive applications, where precision, control, and user experience are critical to task success. However, the integration of diverse devices and the delivery of seamless, high-quality haptic feedback remain significant challenges. Current teleoperation systems often suffer from device-specific limitations and lack interoperability, hindering their adaptability to various operational scenarios [7]. Despite recent advances in teleoperation and immersive robotics, most systems remain limited to specific hardware or isolated user configurations [8]. There is a lack of generalizable, flexible architectures capable of supporting diverse haptic and tracking technologies while enabling collaborative multi-user scenarios [9,10]. This work addresses this gap by proposing and validating a modular, device-agnostic teleoperation system instantiated in four real configurations. Additionally, the traditional focus on network performance metrics, such as latency and throughput, often overlooks the user-centric aspects that define task effectiveness and satisfaction. Addressing these gaps requires a shift toward frameworks that prioritize both the adaptability of teleoperation systems and the comprehensive evaluation of user experience.

### 1.2. QoE in Immersive Teleoperation

In teleoperation, QoE is increasingly recognized as a critical factor for system evaluation [11], extending beyond conventional performance metrics to capture the holistic interaction between users and systems. QoE encompasses both objective and subjective aspects, reflecting the efficiency of task completion and the user’s perception of their experience. For industrial assistive applications, this dual perspective is essential to optimize not only the technical performance but also the usability and satisfaction of operators.

Objective QoE measures [12], such as temporal and spatial efficiency, provide quantifiable data on how well a system supports task execution. Subjective QoE metrics [13], such as presence, engagement, control, sensory integration, and cognitive load, offer insights into how users perceive their interactions and the overall system responsiveness. By integrating these dimensions, immersive teleoperation systems can achieve greater acceptance and effectiveness in real-world applications.

### 1.3. Contributions of This Study

This study addresses the challenges of device-specific teleoperation systems by proposing a collaborative, device-agnostic interface framework designed for smart haptics in immersive environments. By enabling interoperability across diverse haptic feedback devices and tracking technologies, the framework facilitates modular and scalable teleoperation set-ups. This research evaluates the system’s performance in scenarios where users manipulate objects along predefined 3D trajectories, with configurations ranging from one user with a single device configuration (1U-1E) to multiple users with multiple devices (MU-ME).

This article makes several key contributions. First, it presents a comprehensive methodology for evaluating both objective QoE metrics—temporal efficiency and spatial accuracy—and subjective QoE metrics based on user surveys of 23 diverse participants. Second, it demonstrates the feasibility of a device-agnostic teleoperation framework in various industrial scenarios, highlighting the adaptability and performance of different equipment configurations. Third, it explores the relationship between objective task performance and subjective user perceptions, providing valuable insights into designing teleoperation systems that balance technical precision with user satisfaction. These contributions advance the state of the art in immersive teleoperation and establish new benchmarks for evaluating QoE in industrial assistive applications.

### 1.4. Structure of the Document

The remainder of this article is structured as follows. First, Section 2 reviews the related work in immersive teleoperation, haptic feedback technologies, and QoE assessment methodologies, establishing the foundation for the proposed framework. Then, Section 3 describes the system design and architecture, detailing the device-agnostic layer and the integration of haptic devices and trackers. In more detail, Section 4 presents the experimental set-up and methodology, outlining the evaluation metrics and data collection processes. Afterward, Section 5 analyzes the experimental results, comparing the performance of different device configurations and discussing the implications of objective and subjective QoE metrics. In consequence, Section 6 delves into the broader implications of this research, addressing challenges, potential applications, and opportunities for future development. Finally, Section 7 concludes the article with a summary of the findings, contributions to the field, and directions for future work. This structure ensures a systematic and detailed exploration of this study’s objectives, methods, and outcomes, providing a cohesive narrative for advancing immersive teleoperation systems.

## 2. Related Work

The field of immersive teleoperation has witnessed significant advancements in recent years, driven by innovations in robotics [14], haptics [15], and networked communication systems [16]. These developments have enabled systems capable of delivering precise control and realistic feedback to users, even in remote or hazardous environments. However, the diverse landscape of teleoperation technologies presents unique challenges, particularly in achieving interoperability, ensuring real-time responsiveness, and accurately assessing the quality of user experience. This section provides a comprehensive review of the state of the art in immersive teleoperation, focusing on haptic devices, QoE evaluation frameworks, and device-agnostic systems. These reviews establish the context and highlight the gaps that this study seeks to address.

### 2.1. Immersive Teleoperation Systems

Immersive teleoperation systems combine robotic manipulators, haptic interfaces, and visual tracking technologies to enable remote task execution with a high degree of realism. Recent advances [17,18] have focused on integrating multimodal sensory feedback, such as tactile, kinaesthetic, and visual cues, to enhance user engagement and task accuracy. Emerging systems [19,20] leverage mixed reality environments and volumetric rendering to improve spatial awareness and intuitive control. However, most of these systems rely on tightly coupled hardware-software ecosystems, limiting their adaptability to different devices or operational contexts. In particular, the challenge of integrating volumetric visualization with real-time haptic feedback in latency-sensitive applications remains a key bottleneck for scaling immersive teleoperation systems to industrial settings. Despite these advancements, achieving consistent performance across diverse set-ups remains a challenge due to the lack of standardization and modularity. Recent work [21,22] has suggested leveraging distributed architectures and edge computing to offload computationally intensive tasks, such as motion prediction and force rendering, thereby reducing latency and improving responsiveness in complex teleoperation scenarios.

### 2.2. Haptic Feedback Devices and Modalities

Haptic feedback devices [23,24] play a crucial role in immersive teleoperation by providing tactile and kinaesthetic sensations that enhance user control and precision. Grounded devices [25], such as robotic arms and force-feedback mechanisms, deliver high-fidelity responses ideal for precision tasks, while mobile haptic devices, such as vibrotactile gloves and joysticks, offer portability and ease of use. Recent developments in wearable technologies [26], including force-feedback exoskeletons and vibrotactile arrays, have expanded the range of applications, particularly in industrial and medical contexts. However, these devices often require dedicated integration and calibration, which limits their interoperability. An emerging trend in this domain [27,28] is the use of multimodal haptic devices that combine vibrotactile and force-feedback capabilities, enabling more nuanced interaction in teleoperation tasks. For instance, hybrid devices have demonstrated potential for improving task performance in scenarios requiring both gross and fine motor control [8]. Additionally, optimizing the trade-off between fidelity, latency, and power consumption remains an active area of research. Further exploration is needed to establish standardized benchmarks for evaluating these trade-offs across diverse haptic modalities.

### 2.3. Quality of Experience in Robotics and Haptics

The evaluation of user experience in teleoperation has evolved from subjective impressions [29] to structured frameworks that combine objective and subjective metrics [30]. In robotics and haptics, QoE assessments often involve measuring task performance metrics, such as trajectory accuracy and task completion time, alongside user perceptions of control, presence, and cognitive load. Advances in QoE research [31,32] have introduced techniques for real-time monitoring of user interactions, enabling dynamic adjustments to system parameters based on user feedback. Recent studies [33] have emphasized the importance of adaptive QoE models that leverage machine learning to predict user satisfaction based on sensor data and behavioral metrics. These models enable real-time system tuning, which is particularly valuable in collaborative multi-user teleoperation scenarios. Despite this progress, existing studies [34] typically focus on specific devices or configurations, making it challenging to generalize findings to multi-device or device-agnostic systems. Moreover, the interplay between objective and subjective QoE factors remains underexplored, particularly in collaborative and industrial scenarios. The integration of cognitive load estimation, using techniques such as electroencephalogram or eye-tracking data, offers a promising direction for refining QoE assessments in teleoperation tasks.

### 2.4. Device-Agnostic Teleoperation Frameworks

Device-agnostic teleoperation frameworks [35] aim to decouple system functionality from hardware dependencies, enabling the integration of diverse devices within a unified ecosystem. Such frameworks leverage abstraction layers and standardized protocols to support interoperability, scalability, and modularity. Research in this area [36,37] has focused on developing middleware architectures and APIs that facilitate seamless communication between heterogeneous devices. However, achieving real-time performance and maintaining high QoE across varying configurations is a persistent challenge. Recent approaches to device-agnostic frameworks [38,39] have incorporated predictive codecs and context-aware algorithms to optimize communication efficiency, particularly in scenarios involving low-bandwidth or high-latency networks. These techniques have shown promise in improving task completion times and reducing errors in remote teleoperation. While some other studies [40,41] have explored the use of adaptive algorithms and predictive models to optimize performance, the application of these techniques to haptic systems remains limited. Additionally, the lack of comprehensive evaluations in dynamic, multi-user scenarios underscores the need for further investigation into device-agnostic frameworks for collaborative teleoperation.

## 3. System Design and Architecture

The proposed teleoperation system, shown in Figure 2, is built upon a modular and device-agnostic architecture designed to enable seamless integration of diverse haptic feedback and tracking devices. By abstracting device-specific functionalities, the system achieves adaptability and scalability, allowing users to interact effectively in various configurations ranging from single-user set-ups to multi-user collaborative environments. This section elaborates on the structural design of the system, focusing on the core architecture, the device-agnostic layer for interoperability, and the integration of haptic and tracking devices tailored to immersive teleoperation tasks.

### 3.1. Overview of the Teleoperation Framework

The framework adopts a hierarchical design that separates the hardware layer from the control and interaction layers, ensuring that the system can operate independently from specific devices. This separation is achieved through a device-agnostic layer, which provides a standardized interface for communication between hardware components and the central control unit.

To support real-time bidirectional control, the system employs multiple communication protocols across different layers. At the hardware interface level, the UR5e communicates with the OnRobot RG2 gripper via Modbus RTU over USB-serial, while real-time robot-data exchange occurs over RTDE. Within the LCON, core control services run over ROS Noetic, while XML-RPC interfaces expose control commands and sensor status to the RTHN. Inside the RTHN, haptic and tracking control data are marshaled via TCP for tracking and UDP for haptic frames. Audio and video streams originating from the camera and microphone are transmitted via RTSP for session control and RTP for media delivery over the private 5G network. Sensor data from trackers and haptic devices, ranging from USB-serial to BLE interfaces, are received, normalized, and passed through the device abstraction layer, ensuring modular and interoperable integration of heterogeneous devices across the system.

The system also supports multiple operational configurations, including one-user-with-one-equipment (1U-1E), one-user-with-multiple-equipment (1U-ME), and multi-user-with-multiple-equipment (MU-ME). Such flexibility makes the framework suitable for a broad range of industrial and collaborative teleoperation scenarios, enhancing both usability and deployment efficiency. A supplementary video showcasing representative experimental scenarios, synchronized haptic and visual feedback, and real-time user interaction has been made available online [42]. This demonstration complements the figure by illustrating the dynamic behavior of the system in actual deployment.

### 3.2. Device-Agnostic Layer for Interoperability

At the core of the system lies the device-agnostic layer, a middleware architecture designed to abstract hardware functionalities and provide a unified control mechanism, as depicted in Figure 3. This layer leverages standardized communication protocols and adaptable drivers to ensure compatibility across a wide array of devices. Upon initialization, the layer dynamically profiles connected devices, identifying their capabilities and mapping them to the system’s control architecture. Real-time synchronization mechanisms ensure smooth operation even in multi-device set-ups, while modular design principles enable the straightforward integration of new devices.

The device-agnostic layer incorporates advanced scheduling algorithms to optimize resource allocation and manage latency, ensuring that performance remains consistent regardless of the specific hardware configuration. By decoupling hardware-specific tasks from the primary system logic, this layer enhances the scalability and adaptability of the teleoperation framework, enabling seamless transitions between operational modes. To manage concurrency and real-time responsiveness, our middleware uses ROS-level scheduling through “ros::AsyncSpinner”, which spawns multiple threads to process callbacks in parallel across the Content Manager, Capability Profiler, and Event Formatter modules. This ensures that tracking and haptic data streams are handled asynchronously without blocking control flows. Meanwhile, at the lower level, the UR5e controller employs RTDE in a 125 Hz cyclic synchronization mode, coordinating command and status exchanges over TCP/IP. This scheduling strategy aligns input multiplexing and output streaming with the robot’s internal control loop, guaranteeing timely integration and execution of device-agnostic commands.

### 3.3. Integration of Haptic Devices and Trackers

The framework described in Figure 2 is instantiated in four different configurations, each integrating specific haptic feedback and tracking devices to highlight distinct interaction modalities. These configurations share a common architecture shown in Figure 4 while adapting to the unique characteristics of each device configuration.

The configuration-specific tracking control and haptic feedback flows and characteristics are schematized in Figure 5. While the modular pipeline remains consistent across the figures, each diagram highlights the modality-specific components such as feedback encoders (e.g., vibrotactile, force, and gesture), custom preprocessing modules, and synchronization strategies:
**D1: Touch Device (3D Systems).** This grounded kinaesthetic device provides precise positional tracking and high-fidelity force feedback, making it well suited for tasks that demand fine motor control and spatial accuracy. This device connects via USB using the official “TouchDeviceDriver v2023.1.4” and operates as an ROS node. It publishes six-DOF pose and force feedback data over a 1 kHz referesh rate via USB-serial, conforming to standard ROS message types.**D2: DualSense Controller.** This is a tactile joystick that doubles as a mobile haptic feedback device, offering ease of use and portability. Its intuitive interface is ideal for rapid task execution in dynamic environments. It is paired via BLE and managed using the community-supported “ds5_ros” ROS node, built on the “pydualsense” library. It publishes sensor data (axes and buttons) via sensor_msgs and subscribes to feedback topics (rumble and triggers), encapsulating haptic events in the abstraction layer.**D3: bHaptics TactGloves with REalsense Camera Tracker.** This combination of a vibrotactile glove and a camera-based tracking system with video gesture recognition using a convolutional neural network (CNN) across 21 hand landmark positions enables immersive interactions, particularly for one-handed operations requiring tactile feedback. The TactGlove connects via BLE and is driven by the bHaptics “tact-python” library, which publishes haptic commands over ROS topics. The camera is wired via USB and runs an ROS wrapper based on OpenCV and MediaPipe for real-time CNN pose estimation.**D4: SenseGlove Nova with VIVE Trackers.** This combination integrates force-feedback and vibrotactile functionalities with expanded movement tracking capabilities and hand gesture recognition using a supervised feedforward neural network across the string tension of each finger joint, delivering an advanced multi-modal haptic experience. The SenseGlove connects via BLE and uses the SGConnect protocol together with the SGCore API to stream multi-finger force and position data. Tracking data from the Vive Trackers is obtained via USB through the SteamVR subsystem. Both streams are integrated via ROS nodes that publish standardized messages.

In the proposed system, the bHaptics TactSuit X40 (bHaptics Inc., Seoul, Republic of Korea) is employed as a directional passive haptic feedback interface, leveraging its spatially distributed vibrotactile actuators on both the front and back panels to deliver semantic guidance during teleoperation as shown in Figure 6. The front actuators are exclusively mapped to indicate successful path-following behavior, wherein each point of contact along the expected trajectory is dynamically transformed into an actuator index using normalized screen coordinates and a 4×5 grid mapping logic. These indices are activated using a fixed-intensity front-facing vibration pattern. Conversely, back actuators are used to convey error feedback, driven by the direction and magnitude of the deviation vector between the actual and ideal trajectory. The system classifies directional errors into 1 of 15 predefined regions (e.g., upper left or lower right) and activates the corresponding actuator clusters on the back panel, with the intensity scaled to the error norm. This dual-region design ensures a cognitive and spatial separation of feedback semantics, where success is gently reinforced via front haptics and error is emphatically redirected via back haptics, enabling intuitive, non-visual corrective guidance during immersive robotic teleoperation scenarios.

Each device operates within the framework via a unified driver interface, ensuring consistent performance and interoperability. The modular design enables users to swap or combine devices without requiring extensive reconfiguration, making the system highly adaptable to diverse application needs.

## 4. Experimental Design and Methodology

The experimental methodology was carefully designed to evaluate the performance and user experience of the proposed teleoperation system. The experiments focused on measuring both the objective and subjective QoE across diverse configurations of users and equipment. Participants engaged in object manipulation tasks requiring precise trajectory execution, with performance assessed through temporal and spatial metrics. Additionally, subjective perceptions of presence, engagement, control, sensory integration, and cognitive load were collected to gain deeper insights into the user experience. This section describes the experimental set-up, participant demographics, QoE metrics, and data collection methods in detail.

### 4.1. Experimental Set-Up

The experimental set-up was designed to evaluate the efficacy and adaptability of the immersive teleoperation system under varied operational configurations, task complexities, and user-device scenarios. Central to the experimental design was the integration of a device-agnostic framework, enabling seamless operation with multiple haptic and tracking devices. The set-up emphasized reproducibility and consistency by standardizing network parameters and system settings while allowing flexibility to accommodate diverse teleoperation tasks. Data collection was structured to capture a comprehensive range of performance metrics, ensuring a holistic assessment of the system.

#### 4.1.1. System Configuration for XU-XE Scenarios

The experimental design incorporated three distinct operational configurations to evaluate the system’s performance and adaptability under varying user and equipment set-ups, illustrated in Figure 7. Each configuration utilized a specific combination of haptic feedback and tracking devices for real-world teleoperation tasks.

These system configurations are defined as follows:**1U-1E: One User with One Equipment.** In this configuration, a single user operated one device at a time, evaluating its standalone capabilities in performing teleoperation tasks.**1U-ME: One User with Multiple Equipment.** In this configuration, a single user operated two or more devices simultaneously, requiring coordination and adaptability to perform teleoperation tasks.
−**D1 and D2:** The Touch Device provided high-precision manipulation, while the DualSense controller was used for navigation or secondary object placement. This scenario tested the user’s ability to switch focus between detailed and generalized control.−**D3 and D4:** The bHaptics DK2 glove was used for tactile interaction, while the SenseGlove Nova handled force-feedback tasks. This combination assessed the user’s ability to manage and integrate two distinct haptic modalities.**MU-ME: Multiple Users with Multiple Equipment.** This configuration involved multiple users operating separate devices simultaneously to accomplish collaborative tasks. The objective was to evaluate coordination, shared control, and system performance in multi-user scenarios.
−**D1 and D3:** One user employed the Touch Device for precision-based manipulation, while another used the bHaptics DK2 glove for tactile object interaction. This set-up tested complementary task distribution among users.−**D2 and D4:** The DualSense controller was utilized for navigation and coarse manipulation, while the SenseGlove Nova provided force-feedback capabilities for detailed adjustments. This scenario assessed collaborative efficiency in handling complex objects and trajectories.

Across all configurations, the device-agnostic layer enabled seamless interoperability, ensuring that each device could be integrated without requiring extensive reconfiguration.

#### 4.1.2. Object Manipulation and 3D Trajectory Tasks

Participants performed tasks involving the manipulation of predefined objects—regular geometric shapes such as cubes, spheres, and cylinders—placing them into designated baskets along a predefined 3D trajectory. The trajectory varied in complexity from linear paths to curved and multi-turn routes, as depicted in Figure 8, simulating real-world teleoperation scenarios.

The system recorded user movements in real time, capturing deviations from the intended trajectory and task completion times. These tasks were chosen to challenge both the haptic and tracking components of the system while requiring a combination of precision and control from the participants.

In addition to predefined static trajectories, a subset of scenarios introduced mild dynamic variability by simulating slight object displacement and orientation changes at runtime. These variations required participants to adjust grasping angles or apply minor trajectory corrections in real time. While the workspace layout remained constant, this variability aimed to mimic industrial settings, where environmental changes occur due to machine operation or external disturbances.

### 4.2. Participant Demographics and Selection

The study recruited 23 participants from diverse demographics, ensuring a representative sample in terms of age, gender, educational background, and familiarity with technology (Figure 9). Participants were grouped based on their prior experience with haptic devices, robotics, and virtual environments to analyze the impact of these factors on performance and user experience.

In particular, they were categorized into three groups based on their self-referred skill levels:**Novice Users:** Individuals with little or no prior experience with teleoperation or haptic devices;**Intermediate Users:** Participants with moderate familiarity, such as casual gamers or those with minimal robotics exposure;**Expert Users:** Professionals or researchers with significant experience in robotics, haptics, or related technologies.

The distribution of participants across demographic and skill-related categories was designed to maintain balance and diversity. Specifically, 26% of the participants belonged to AG1, 39% belonged to AG2, and 35% belonged to AG3. Regarding teleoperation skill levels, participants were classified as follows: 35% TS1, 39% TS2, and 26% TS3. Gender representation was also balanced, with 57% identifying as male and 43% identifying as female. Finally, for the study level of the participants, 22% were classified as SL1, and 78% were classified as SL2. These proportions ensured a heterogeneous sample, allowing for robust subgroup analysis of performance and experience metrics.

Furthermore, the inclusion criteria ensured that participants had no significant physical impairments that could affect their ability to interact with the haptic devices or perform the required tasks. Pre-experiment questionnaires collected detailed demographic data, which was later used to identify correlations between the participant characteristics and QoE metrics.

### 4.3. Objective Performance Metrics

Objective performance metrics were designed to quantitatively evaluate the operational efficacy of the teleoperation system. These metrics encompassed two primary dimensions: temporal efficiency, reflecting task execution speed, and spatial accuracy, indicating adherence to predefined trajectories. Together, they offered a comprehensive measure of system performance, user interaction quality, and device functionality.

Temporal efficiency ($T_{E}$) was defined as the average time required by participants to complete a given task over three optional tries, from initial object manipulation to placement along the prescribed trajectory. This metric directly assessed the responsiveness of the system and the user’s ability to operate efficiently. This metric aligns with established teleoperation studies that link quicker task execution to enhanced performance [43,44]. Completion times were recorded for each XU-XE configuration and trajectory complexity, with statistical analyses performed to determine means, variances, and potential performance bottlenecks.

Spatial accuracy ($S_{A}$) was assessed by quantifying the alignment between the user-executed trajectory and the predefined ideal path. This approach is widely adopted in remote manipulation and VR interaction research to quantify precision and control effectiveness [45,46]. Metrics such as the mean deviation (average distance between user trajectory and reference path), cumulative error (sum of all deviations along the trajectory), and path smoothness (consistency of movement) were calculated to characterize accuracy.

### 4.4. Subjective Experience Metrics

Subjective experience metrics were employed to capture the participants’ perceptions and qualitative feedback on their interaction with the teleoperation system. These metrics provided insights into the psychological and sensory aspects of the user experience, offering a valuable complement to the objective performance data. The evaluation focused on five core dimensions:**Presence:** Commonly referred to as “telepresence”, this is defined in the literature as the user’s illusion of being physically present in a mediated environment, an experiential phenomenon where the medium itself “disappears” from awareness [47].**Engagement:** Defined as the degree of psychological investment, focus, and enjoyment a user experiences during a task [48], we assessed engagement through self-reporting and behavioral indicators like persistence and error recovery.**Control:** This refers to the user’s perceived command over the teleoperation system, encompassing confidence, responsiveness, and synchronization of actions and feedback [49].**Sensory integration:** This denotes the cohesion of multisensory cues—visual, haptic, and proprioceptive—and their combined effect on perceptual realism [48].**Cognitive load:** This evaluates the mental effort and mental resources required to operate the system, particularly in regard to the effort needed to manage sensory input and control complexity [50].

### 4.5. Data Collection and Analysis Techniques

Data collection employed a multi-modal approach, combining system-generated logs, video recordings, and participant feedback from individual experiments, as exemplified in Figure 10.

Objective metrics, such as completion times and trajectory deviations, were extracted from system logs and analyzed using statistical methods to identify patterns and correlations. Subjective metrics were collected through standardized questionnaires, including Likert 1–5 scales and open-ended questions, to capture nuanced user experiences. The following questions were used along the five core dimensions introduced in Section 4.4:To what extent did you feel immersed in the remote environment during the task?Did haptic feedback make you feel as if you were physically interacting with objects?How focused and attentive did you feel during the task?Did you enjoy the teleoperation experience using this configuration?How confident were you in your ability to control the robotic system?Did the system respond accurately and promptly to your commands?How naturally did the visual, haptic, and proprioceptive cues combine during tasks?Were you able to easily interpret the feedback from the device(s)?How mentally demanding did you find the task?Did you need to frequently pause or think carefully to continue the operation?

Regarding ethics considerations, no personally identifiable information (e.g., name, email, or contact details) was collected. Participant responses were anonymized and analyzed exclusively in aggregated form. This study complied with the data protection standards outlined by the European Union under the GDPR framework. All data were securely stored on GDPR-compliant servers within the EU, with access restricted to authorized researchers of the TOAST Doctoral Network (https://www.toast-dn.eu), ensuring complete confidentiality.

The data analysis process involved statistical testing to compare results across configurations and participant groups, providing both quantitative and qualitative insights into the system’s performance and user experience. The SC7 and SC8 scenarios introduced higher visual and haptic feedback density with rapid context switching, contributing to increased an mental workload and cognitive overflow risk, especially for untrained users, which justified optional wearable instrumentation for those cases.

## 5. Results and Analysis

This section provides a dual-layered exploration of system performance, encompassing both objective metrics and subjective user experiences. The interplay between these dimensions forms the basis for understanding how various device configurations, task complexities, and interaction modalities influence teleoperation efficacy. The analysis draws on a comprehensive dataset derived from experiments with diverse set-ups, offering insights into temporal and spatial performance, user perception, and cognitive load.

### 5.1. Objective Performance Results

The results of the objective performance metrics provided a detailed understanding of the operational dynamics across device configurations. In the following, we delve into the nuances of how these metrics varied across configurations and experimental conditions, contributing to a comprehensive evaluation of system efficacy.

As a reference point for evaluating user-driven teleoperation performance, a baseline benchmark was established using fully scripted trajectory executions performed by the system without human input. These benchmark runs leveraged pre-programmed actuator paths and ideal motion planning without user control loops or feedback latency. The resulting average task completion time ($T_{C}^{\mathrm{ideal}}$) was 4.2 s, and the trajectory similarity ($S_{T}^{\mathrm{ideal}}$) consistently exceeded 0.98, indicating near-perfect adherence to the target path.

This baseline establishes a theoretical upper bound for system efficiency under perfect conditions. Comparing this against the user-driven metrics allowed us to quantify human-in-the-loop performance gaps and identify which device configurations approached ideal responsiveness and precision. Notably, D4 and D1 in the 1U-1E configuration achieved mean $T_{C}$ values within 40–50% of the baseline and $S_{T}$ values above 0.89, demonstrating high teleoperation fidelity. Conversely, D2 and D3 in multi-device or asynchronous set-ups showed a larger deviation from the benchmark, suggesting a need for improved feedback coherence and latency reduction.

During experiments, system-level performance was also evaluated to ensure real-time responsiveness. The end-to-end latency for haptic feedback averaged 65 ms when operating over the 5G standalone private network, while the video stream exhibited latencies ranging from 150 to 290 ms, depending on the resolution and encoder settings. The bidirectional throughput during full operation in the MU-ME configuration reached up to 25 Mbps, establishing a preliminary requirement for synchronized immersive teleoperation.

#### 5.1.1. Temporal Efficiency Across Device Configurations

The temporal efficiency results shown in Figure 11 underscored the interplay between device design and task complexity. The observed variations in task completion time ($T_{C}$) reflected differences in system responsiveness, user familiarity, and task-specific requirements.

D1 consistently achieved the shortest task completion times for precision-dependent tasks, leveraging its grounded kinaesthetic feedback to facilitate intuitive control. D4 exhibited comparable performance, particularly in tasks with dynamic trajectories, where its robust force-feedback system proved advantageous. In contrast, D2 displayed efficiency in simpler paths but showed notable delays in tasks requiring complex spatial navigation, as the joystick mechanism introduced mechanical limitations. D3 exhibited the longest completion times, attributed to intermittent latency in its external tracking system, despite its vibrotactile feedback providing moderate guidance.

The impact of configuration was also evident, with the MU-ME scenarios introducing the highest $T_{C}$ values. This configuration required participants to coordinate actions across multiple devices, resulting in increased cognitive demand and synchronization challenges. Conversely, 1U-1E set-ups yielded the most efficient results due to the simplified interaction paradigm.

#### 5.1.2. Trajectory Similarity Across Scenarios

Trajectory similarity ($S_{T}$) analysis, shown in Figure 12, provided critical insights into spatial performance, quantifying the precision with which users adhered to predefined 3D paths. The results showed significant variability in $S_{T}$ across devices, configurations, and task complexities. Notably, older participants and those with intermediate skill levels exhibited lower $S_{T}$ scores across all devices, particularly with D2 and D3, suggesting a steeper learning curve or less intuitive control mapping for these groups.

D4 exhibited the highest $S_{T}$ scores, achieving mean similarity values above 0.92 across all scenarios. Its combination of force feedback and high-fidelity tracking allowed users to maintain precise alignment with the intended paths. D1 followed closely, benefiting from its grounded interface, which offered stability and control, particularly in tasks requiring static positioning or minimal trajectory deviation.

D2 and D3 demonstrated lower trajectory similarity, with mean $S_{T}$ scores of 0.78 and 0.74, respectively. For D2, this was attributed to the limitations of its joystick interface, which lacked the fine-tuned precision necessary for complex paths. D3’s performance was hindered by tracking irregularities, which occasionally caused users to overshoot or deviate from the intended path.

In summary, the objective performance results highlight the distinct advantages and limitations of each device. While D1 and D4 excelled in terms of both temporal and spatial metrics, the findings for D2 and D3 point to areas for enhancement, such as reducing latency and improving control fidelity. The analysis provides actionable insights for refining teleoperation systems to optimize performance across diverse application domains.

### 5.2. Subjective Experience Results

The subjective experience analysis offered a nuanced understanding of how participants perceived and interacted with the teleoperation system across configurations.

#### 5.2.1. User Feedback on Presence and Engagement

Participants consistently reported varying levels of presence, defined as the feeling of “being there” within the remote environment, across device configurations (Figure 13).

D4 achieved the highest presence scores, attributed to its realistic force-feedback system and immersive sensory experience provided by the VIVE trackers. Users frequently described the tactile sensations as “intuitive” and “natural”, contributing to a stronger sense of spatial awareness and embodiment.

D1 also performed well in this dimension, leveraging its kinaesthetic feedback to provide a grounded interaction experience, particularly in tasks requiring precision. However, the presence scores for D2 and D3 were notably lower. D2’s tactile joystick design was described as “mechanical” and less immersive, while D3 faced challenges due to intermittent tracking delays, which disrupted the flow of interaction and reduced the sense of immersion.

The engagement results mirrored those for presence, with D4 and D1 outperforming the other devices. Participants noted that the dynamic and responsive feedback provided by these devices made tasks more enjoyable and immersive. In contrast, D2 and D3 showed a decline in engagement, particularly in scenarios requiring repetitive error correction or adjustments. Observational data supported these findings, with higher task abandonment rates in the D2 and D3 configurations during complex scenarios.

#### 5.2.2. Insights into Sensory Integration and Control

The sensory integration analysis, shown in Figure 14, highlighted the critical role of cohesive feedback in enhancing user performance. Participants reported the highest integration scores with D4, where visual, haptic, and proprioceptive feedback seamlessly merged, creating a cohesive operational experience. This was particularly evident in tasks requiring simultaneous visual tracking and precise manipulation, where users described the feedback as “synchronized” and “intuitive”.

D1 also demonstrated strong sensory integration, particularly in scenarios emphasizing kinaesthetic feedback. However, D2 and D3 scored lower in this dimension. For D2, participants reported a disconnect between the visual and tactile feedback, describing the interaction as “fragmented”. D3 faced similar challenges, with the vibrotactile glove and external camera tracker sometimes providing asynchronous signals, resulting in a less natural user experience.

The control metrics revealed a similar trend. The D4 users expressed high confidence in their ability to execute precise movements, with the force-feedback glove providing tactile resistance that enhanced their perception of control. The D1 users also reported a strong sense of control, particularly in static positioning tasks. Conversely, the D2 and D3 participants noted difficulty in achieving fine motor precision, with some describing the controls as “laggy” or “imprecise”.

#### 5.2.3. Cognitive Load Variations Across Devices

Cognitive load analysis provided valuable insights into the mental effort required to operate each device configuration, as shown in Figure 15.

D4 consistently demonstrated the lowest cognitive load, as its intuitive feedback mechanisms and precise tracking reduced the need for error correction and mental adjustments. Participants frequently described their interactions as “effortless” and “smooth”, even in high-complexity tasks.

D1 followed, with users citing the grounded interface as reducing the cognitive demands associated with spatial positioning and manipulation. In contrast, D2 and D3 exhibited higher cognitive load scores. D2 users reported difficulties adapting to the joystick interface, which required significant mental effort to translate input into precise actions. The D3 users noted that tracking inconsistencies increased their need to constantly adjust and recalibrate, leading to higher mental fatigue.

Regression analysis of cognitive load against task complexity further emphasized the differences among devices. While D4 and D1 maintained relatively stable load levels across tasks, D2 and D3 showed steep increases in cognitive demand as the path intricacy grew. These results underscore the importance of designing systems that minimize cognitive effort, particularly for applications requiring sustained user engagement.

In summary, the subjective experience results underscore the variability in user perception across devices. While D4 and D1 consistently provided immersive, engaging, and intuitive experiences, the findings for D2 and D3 highlight opportunities for improvement, particularly in feedback synchronization and control mechanisms. These insights are critical to refining the design of teleoperation systems to prioritize user-centric performance.

### 5.3. Comparative Analysis of Configurations

Next, we provide a comparative analysis of system configurations for a comprehensive understanding of trade-offs between performance and usability across experimental set-ups. By integrating objective metrics with subjective feedback, as depicted in Figure 16, this section synthesizes the strengths and limitations of the XU-XE configurations, offering actionable insights for future system optimization.

#### 5.3.1. Performance Trade-Offs Between Configurations

The 1U-1E configuration, where a single user operated a single device, demonstrated a consistent balance between temporal efficiency and spatial accuracy. Devices D1 and D4 excelled in this set-up, with participants achieving high trajectory similarity and low task completion times. This configuration highlighted the advantage of minimizing system complexity, as users could focus solely on mastering one device’s interaction paradigm. However, subjective feedback indicated that prolonged use of D2 and D3 in this set-up led to fatigue due to higher cognitive demands, particularly in high-complexity tasks.

In contrast, the 1U-ME configuration, where a single user operated multiple devices, introduced significant challenges in terms of coordination and cognitive load. While the combined use of D4 and D1 provided complementary feedback mechanisms that improved spatial accuracy in intricate trajectories, participants often reported difficulty in managing the simultaneous input channels. For instance, the integration of D3’s camera tracker with D1’s grounded feedback required users to mentally reconcile disparate sensory inputs, resulting in slower task execution and lower subjective control ratings.

The MU-ME configuration, involving multiple users with multiple devices, presented a unique set of trade-offs. Collaboration between users operating D4 and D2 often resulted in improved temporal efficiency due to distributed task responsibilities. However, this set-up also revealed bottlenecks in communication and synchronization, particularly when users relied on asynchronous feedback devices such as D3. Despite these challenges, the collaborative nature of the MU-ME configuration significantly enhanced subjective engagement and presence, as the participants valued the shared task dynamic.

#### 5.3.2. Correlations Between Objective and Subjective QoE

The interplay between objective and subjective QoE metrics offered critical insights into the holistic performance of the teleoperation system. Statistical analysis revealed strong correlations between the spatial accuracy and subjective control ratings, particularly for D4 and D1. Users who achieved higher trajectory similarity consistently reported a stronger sense of control and lower cognitive load, suggesting that precise feedback mechanisms directly enhance user confidence and task efficiency.

Temporal efficiency, while less strongly correlated with subjective metrics, showed a significant impact on the engagement scores. Faster task completion times, particularly in the 1U-1E configuration with D4, were associated with higher reported enjoyment and focus levels. Conversely, configurations involving D2 and D3, which exhibited greater variability in completion times, often led to frustration and decreased engagement.

The most notable finding emerged from the analysis of sensory integration and presence. Devices that provided cohesive feedback across multiple sensory modalities, such as D4 in the 1U-1E and MU-ME set-ups, consistently scored highest in both dimensions. In contrast, D3’s asynchronous feedback mechanisms highlighted the detrimental impact of poorly integrated sensory inputs, as users reported decreased immersion and increased cognitive load.

These findings underscore the importance of aligning objective performance metrics with subjective user experience to design teleoperation systems that are both efficient and user-centric. By prioritizing cohesive feedback mechanisms and minimizing system-induced cognitive demands, future configurations can achieve a more seamless and intuitive operational experience. This comparative analysis provides a robust foundation for refining device-agnostic interfaces and advancing the field of immersive teleoperation.

## 6. Discussion

This section synthesizes the results, offering critical insights into the implications of device-agnostic teleoperation systems, their optimization using quality of experience (QoE) metrics, and the inherent challenges in balancing objective and subjective performance. Additionally, the practical applications of these findings in industrial and assistive contexts are explored.

### 6.1. Implications for Device-Agnostic Teleoperation

The results underscore the potential of device-agnostic teleoperation systems in enhancing flexibility and scalability for diverse applications. The seamless integration of heterogeneous devices, such as the Touch Device, DualSense controller, bHaptics TactGlove, and SenseGlove Nova, demonstrates that a robust device-agnostic layer can standardize communication and interaction protocols. This capability not only reduces the system configuration overhead but also enables dynamic adaptation to user and task-specific needs. Our experiments revealed that, despite hardware-level differences, all four devices achieved comparable performance in terms of both spatial accuracy and temporal efficiency metrics. These outcome similarities support the practical viability of a device-agnostic teleoperation framework. While the SenseGlove Nova and Touch Device performed slightly better in tasks requiring fine motor control due to richer kinesthetic feedback, the DualSense and bHaptics DK2 still enabled consistent task completion. These results indicate that standardized abstraction layers and harmonized message formats can effectively mask low-level differences, validating our approach beyond theoretical expectations.

However, the study also highlighted that the effectiveness of a device-agnostic framework depends not only on abstraction but on the synchronization and integration of multimodal sensory feedback. Devices like the bHaptics TactGlove, which offer less synchronized feedback, induced a slightly higher cognitive load, even though task performance remained within the same statistical range. Devices that provide synchronous and multimodal feedback, such as the SenseGlove Nova, are better suited for these systems, as they minimize cognitive friction and enhance user control. Conversely, asynchronous feedback mechanisms, as observed with the bHaptics TactGlove, can undermine user experience, emphasizing the need for optimized synchronization protocols.

### 6.2. Optimizing Robotics Learning Through QoE Metrics

The dual analysis of temporal efficiency and spatial accuracy alongside subjective QoE metrics offers a novel approach to refining teleoperation systems for robotics learning. Objective metrics provide concrete benchmarks for system performance, while subjective assessments capture the nuanced human factors influencing usability.

The correlation between high spatial accuracy and positive user experiences, particularly in tasks involving intricate trajectories, highlights the importance of precision in robotics learning. Systems that prioritize spatial fidelity not only improve task success rates but also foster user confidence, accelerating the learning curve for new operators. Additionally, the emphasis on cognitive load as a subjective metric provides valuable insights into user adaptability, enabling the design of systems that accommodate varying expertise levels.

By leveraging these metrics, future teleoperation systems can adopt an iterative improvement approach, using data-driven insights to enhance both hardware configurations and training methodologies.

### 6.3. Challenges in Balancing Objective and Subjective QoE

Achieving an optimal balance between objective performance and subjective user satisfaction remains a critical challenge in teleoperation system design. While objective metrics such as the task completion time and trajectory similarity offer quantifiable indicators of system efficiency, they may not always align with the subjective experiences of users. For instance, configurations that maximize temporal efficiency may inadvertently increase cognitive load, diminishing user satisfaction.

This dichotomy underscores the need for a holistic evaluation framework that considers trade-offs between efficiency and usability. Future research must explore adaptive interfaces that dynamically adjust feedback mechanisms based on real-time assessments of user performance and cognitive state. Such systems could leverage machine learning algorithms to predict and mitigate user fatigue, ensuring a seamless operational experience across diverse scenarios.

### 6.4. Applications in Industrial and Assistive Scenarios

The findings of this study have significant implications for the deployment of teleoperation systems in industrial and assistive contexts. In industrial applications, the ability to integrate device-agnostic configurations enhances operational versatility, allowing workers to seamlessly switch between tasks and tools. The high spatial accuracy observed with devices like the Touch Device and SenseGlove Nova is particularly advantageous for precision-based tasks such as assembly, inspection, and remote maintenance.

In assistive scenarios, the focus on subjective QoE metrics aligns with the goal of creating intuitive and user-friendly systems for individuals with varying levels of physical and cognitive ability. Devices that combine kinesthetic and vibrotactile feedback, such as the DualSense controller, can provide a tailored user experience, enabling effective interaction with robotic aids. Moreover, the collaborative potential of MU-ME configurations opens new avenues for shared control in caregiving and rehabilitation, fostering improved outcomes through coordinated efforts.

By addressing the challenges and opportunities outlined in this discussion, teleoperation systems can advance toward greater adaptability, efficiency, and inclusivity, paving the way for broader adoption in critical applications.

## 7. Conclusions and Future Work

This section summarizes the key findings, highlights the contributions to robotics and haptics research, acknowledges the limitations of the study, and outlines promising directions for future exploration in immersive teleoperation systems.

### 7.1. Summary of Findings

This study investigated the integration of device-agnostic teleoperation systems, emphasizing the interplay between objective performance and subjective user experience across diverse configurations. The key results demonstrated the strengths of single-device set-ups (1U-1E) in achieving high spatial accuracy and low cognitive load, while multi-device configurations (1U-ME and MU-ME) highlighted challenges related to sensory integration and coordination.

The objective metrics revealed that devices offering precise kinesthetic feedback, such as the Touch Device and SenseGlove Nova, significantly enhanced the trajectory similarity, particularly in complex tasks. Temporal efficiency was highest in scenarios leveraging synchronous feedback mechanisms, underscoring the importance of minimizing system latency. The subjective metrics highlighted the critical roles of presence, control, and cognitive load in shaping user satisfaction, with multimodal feedback proving essential for an immersive experience.

### 7.2. Contributions to Robotics and Haptics Research

This research makes significant contributions to the fields of robotics and haptics by presenting a comprehensive evaluation of device-agnostic teleoperation systems. It provides a robust framework for assessing both objective and subjective quality of experience (QoE) metrics, offering actionable insights into the design of user-centric systems. This study also emphasizes the importance of integrating multimodal sensory feedback and optimizing device synchronization to enhance both performance and usability.

Furthermore, the comparative analysis of configurations establishes a nuanced understanding of the trade-offs inherent in teleoperation set-ups. This contributes to the ongoing discourse on balancing operational efficiency with user experience, paving the way for more adaptable and scalable systems in industrial and assistive applications.

### 7.3. Limitations of the Study

Despite its contributions, this study has several limitations that warrant acknowledgment. First, the scope of testing was constrained to a limited set of devices and configurations, which may not fully capture the diversity of available haptic and tracking technologies. Additionally, the experimental tasks focused on object manipulation along predefined trajectories, which may not generalize to other teleoperation contexts requiring dynamic decision making or fine motor control.

The participant pool, while diverse, was limited in size, potentially introducing biases in subjective feedback. Furthermore, external factors such as environmental conditions and prior user experience with specific devices were not systematically controlled, which could have influenced the results. Finally, while the device-agnostic layer was effective in integrating heterogeneous systems, further validation is needed in real-world deployments to assess scalability and robustness.

### 7.4. Directions for Future Research

Future research should address the limitations identified in this study by expanding the range of devices and teleoperation scenarios. Investigating dynamic and unstructured tasks, such as collaborative assembly or real-time problem-solving, could provide deeper insights into the adaptability of device-agnostic systems. Additionally, longitudinal studies with larger and more diverse participant groups are necessary to evaluate long-term usability and learning curves.

The development of advanced multimodal feedback mechanisms, including the integration of auditory cues or thermal feedback, presents another promising avenue. Combining these modalities with machine learning algorithms to adapt feedback in real time could further enhance both performance and user satisfaction.

Finally, exploring applications in complex industrial environments and healthcare settings, such as robotic-assisted surgery or remote maintenance of hazardous systems, will be crucial for validating the practical utility of device-agnostic teleoperation systems. By addressing these areas, future research can contribute to the realization of more intuitive, efficient, and inclusive teleoperation technologies.

## Figures and Tables

**Figure 1 sensors-25-03993-f001:**
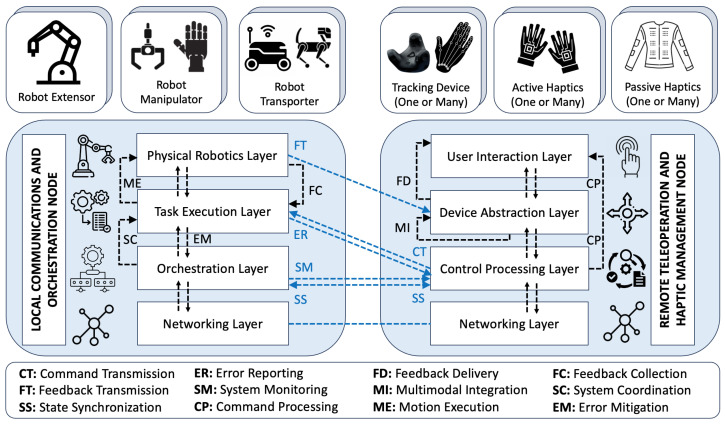
Layered architecture of the immersive teleoperation system, divided into the Remote Teleoperation and Haptic Management Node (RTHN) and the Local Communications and Orchestration Node (LCON). The RTHN layers process user inputs and deliver multimodal feedback through haptic and tracking devices, ensuring seamless control over the robotic system. The LCON layers coordinate task execution, integrating commands from the RTHN for precise and dynamic operation. Inter-layer interactions facilitate real-time communication, while high-level orchestration ensures task efficiency and system synchronization.

**Figure 2 sensors-25-03993-f002:**
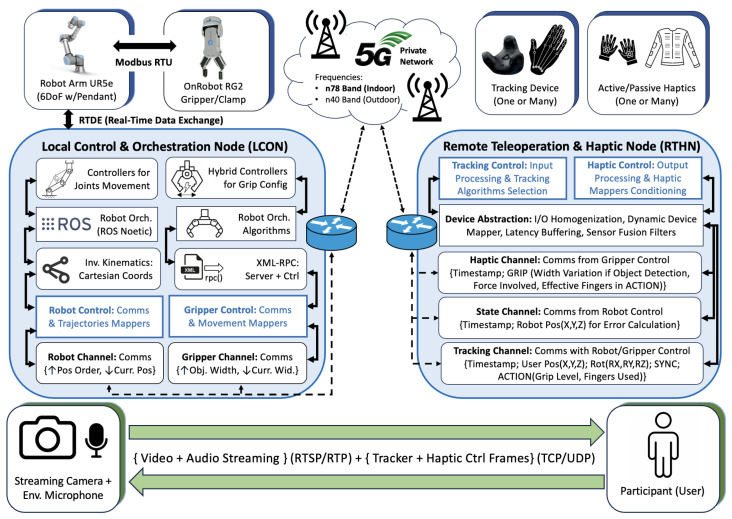
System architecture used for experiments on immersive bilateral teleoperation considering audio, video, and haptic data exchange between nodes. The architecture supports real-time tracking, device abstraction, and haptic feedback through modular channels over a 5G private network.

**Figure 3 sensors-25-03993-f003:**
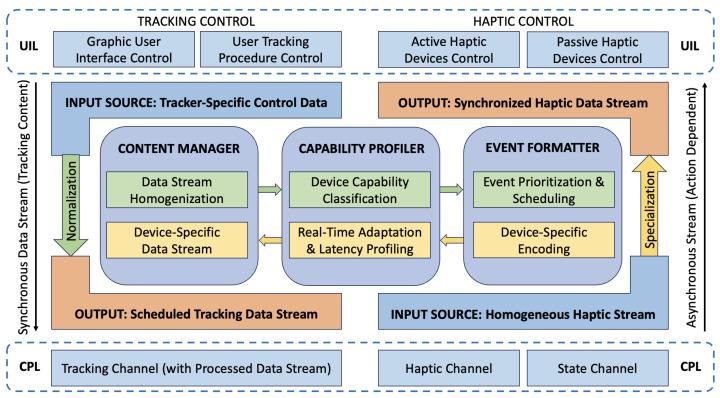
Bidirectional functional architecture of the device abstraction Layer, depicting the dual pipelines for (**left**) tracking input normalization and (**right**) haptic output specialization. The core components—Content Manager, Capability Profiler, and Event Formatter—ensure real-time synchronization, dynamic profiling, and abstraction of heterogeneous tracking and haptic devices. Data streams are routed from the user interaction layer (UIL) to the control processing layer (CPL) to ensure modular and low-latency communication.

**Figure 4 sensors-25-03993-f004:**
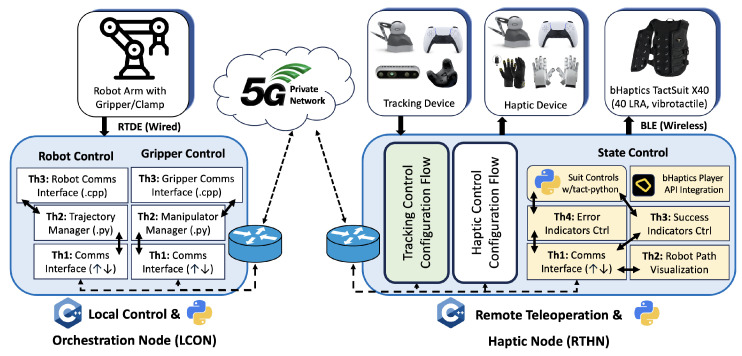
Common modules of the teleoperation framework showing the robot and gripper control components of the LCON and the state control elements of the RTHN, integrating UR robot execution and bHaptics TactSuit X40 with standard tracking and haptic control procedures.

**Figure 5 sensors-25-03993-f005:**
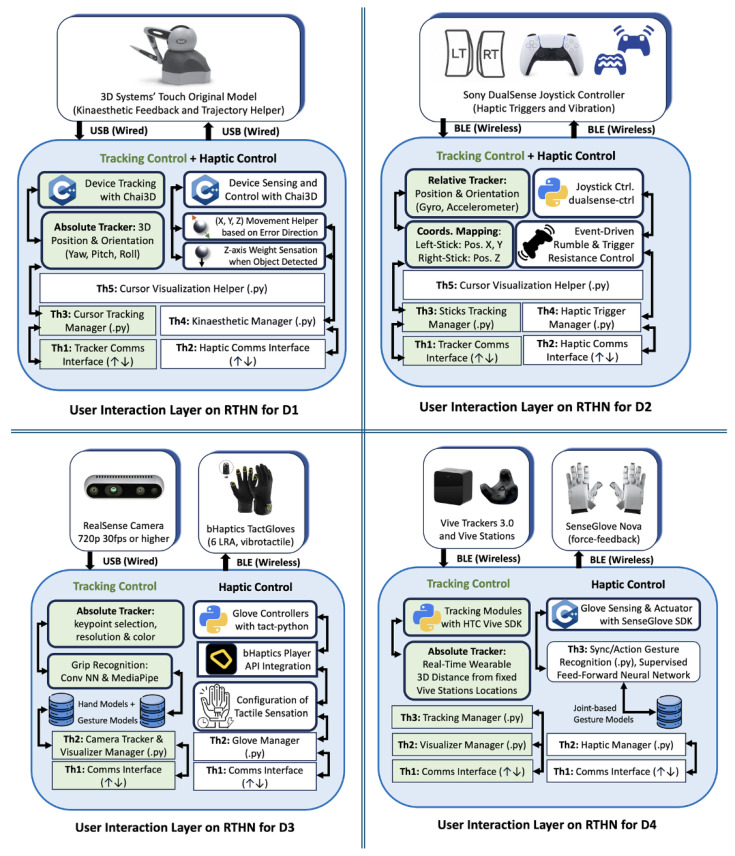
User interaction layers for the RTHN in diverse device configurations. **Top Left (D1)**: Touch Device. **Top Right (D2)**: DualSense controller. **Bottom Left (D3)**: bHaptics TactGloves with RealSense camera. **Bottom Right (D4)**: SenseGlove Nova with Vive Trackers.

**Figure 6 sensors-25-03993-f006:**
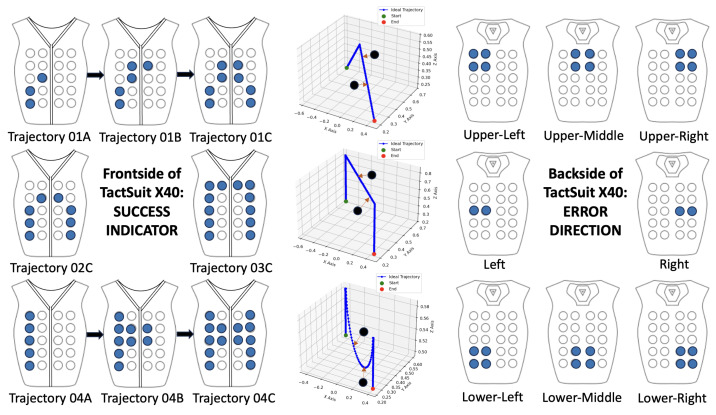
Layouts of the non-simultaneous success and error direction Indicators, updated each 500 ms. The success indicator consists of constant frontal vibration enabled when a success position near the objective trajectory is reached. The error indicator consists of blinking back vibration according to the error direction enabled when not in a success position.

**Figure 7 sensors-25-03993-f007:**
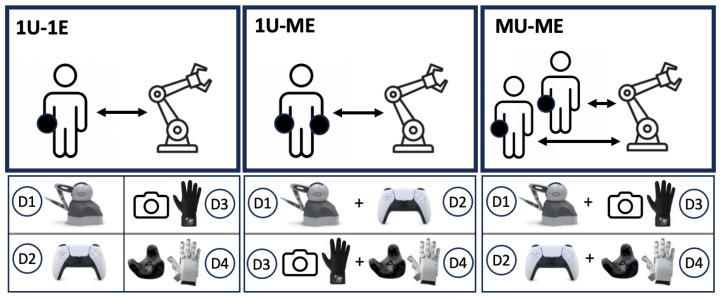
System configurations illustrating one-user-one-equipment (1U-1E), one-user-multiple-equipment (1U-ME), and multiple-users-multiple-equipment (MU-ME) set-ups.

**Figure 8 sensors-25-03993-f008:**
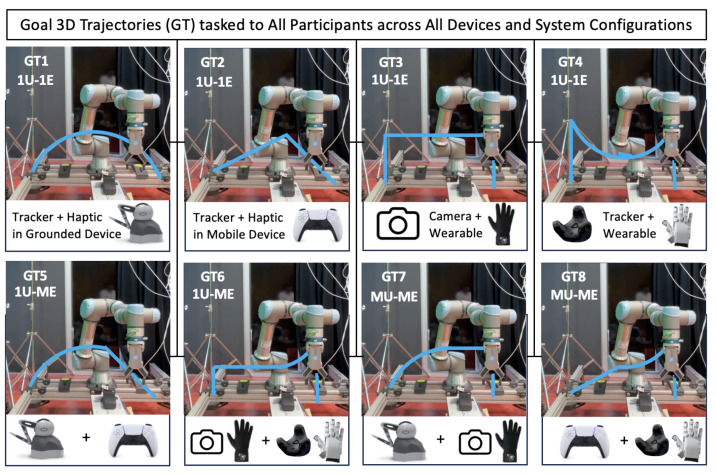
Predefined 3D trajectories (blue lines) assigned to participants across eight scenarios (SC1–SC8). Each trajectory required precise manipulation of the robotic arm to follow specific spatial paths involving varying levels of curvature, complexity, and alignment challenges.

**Figure 9 sensors-25-03993-f009:**
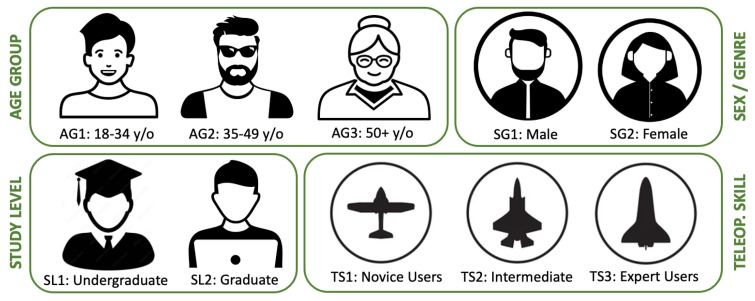
Pariticipant characterizations based on four key demographic and skill-related attributes: age group (AG), sex or gender (SG), study level (SL), and teleoperation skill (TS).

**Figure 10 sensors-25-03993-f010:**
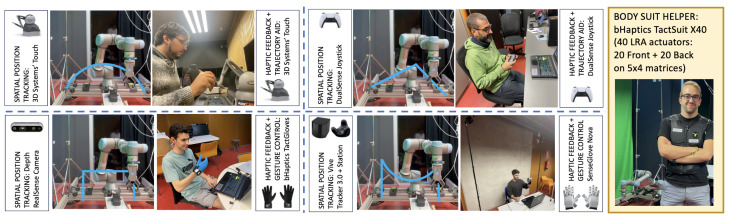
Participants performing teleoperation tasks with specific system configurations and goal trajectories. Usage of a body suit was optional for some sensitive cases, as requested by users with risk of cognitive overflow, particularly for the SC7 and SC8 scenarios, where increased interface complexity and sensory load introduced a heightened risk of cognitive overflow.

**Figure 11 sensors-25-03993-f011:**
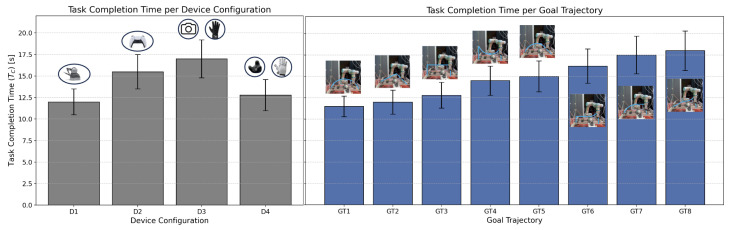
Task completion time analysis across device configurations (D1–D4), system configurations (1U-1E, 1U-ME, and MU-ME), and goal 3D trajectories (GT1–GT8).

**Figure 12 sensors-25-03993-f012:**
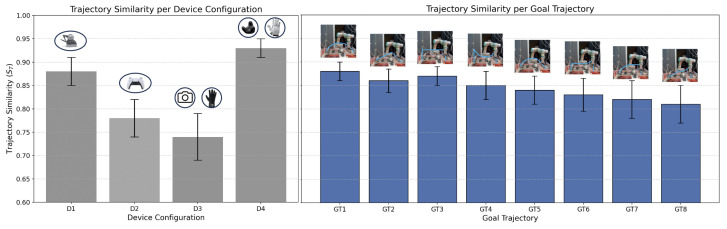
Trajectory similarity analysis across device configurations (D1–D4), system configurations (1U-1E, 1U-ME, and MU-ME), and goal 3D trajectories (GT1–GT8).

**Figure 13 sensors-25-03993-f013:**
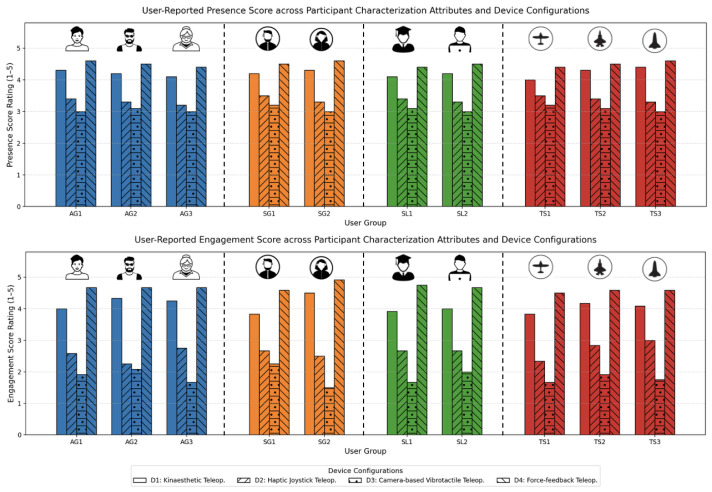
User-reported presence (**top**) and engagement (**bottom**) ratings across all participant characterization attributes and device configurations. D4 consistently received the highest ratings for both presence and engagement, followed by D1. In contrast, D2 and D3 showed lower scores, particularly among older users (AG3), intermediate-skilled users (TS2), and graduates (SL2).

**Figure 14 sensors-25-03993-f014:**
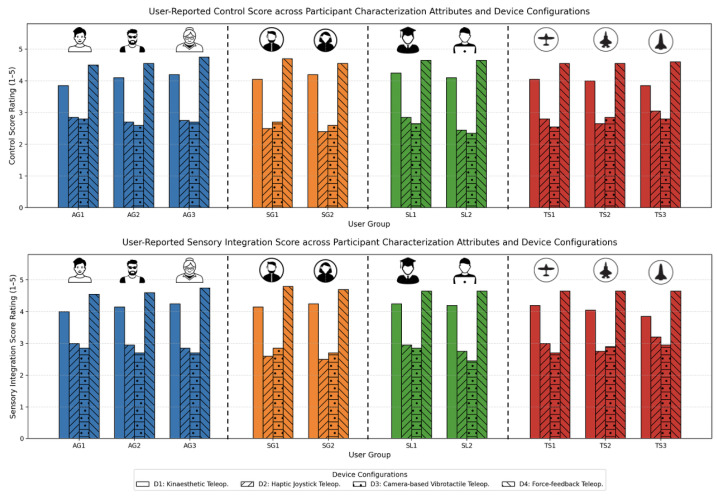
User-reported control (**top**) and sensory integration (**bottom**) scores across participant characterization attributes and device configurations. D4 consistently yielded the highest ratings in both metrics, benefiting from its immersive force-feedback and multisensory coherence. D1 followed with strong scores, especially in tasks involving kinaesthetic precision. In contrast, D2 and D3 demonstrated lower scores, revealing challenges in sensory synchronization and control fidelity, particularly among novice users or in high-complexity tasks.

**Figure 15 sensors-25-03993-f015:**
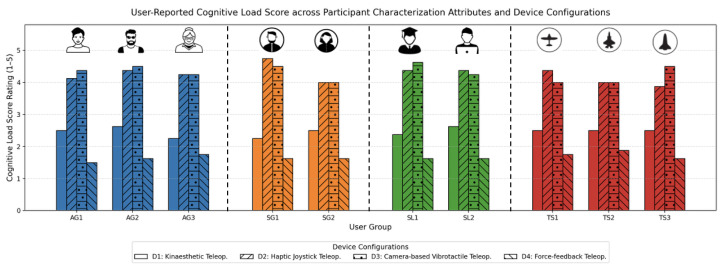
User-reported cognitive load ratings across participant characterization attributes and device configurations. The results reflect higher perceived mental effort for D2 and D3, particularly among users with limited experience or higher task complexity. Conversely, D4 and D1 consistently scored lower for cognitive load, suggesting that intuitive feedback and grounded interaction design help reduce user strain.

**Figure 16 sensors-25-03993-f016:**
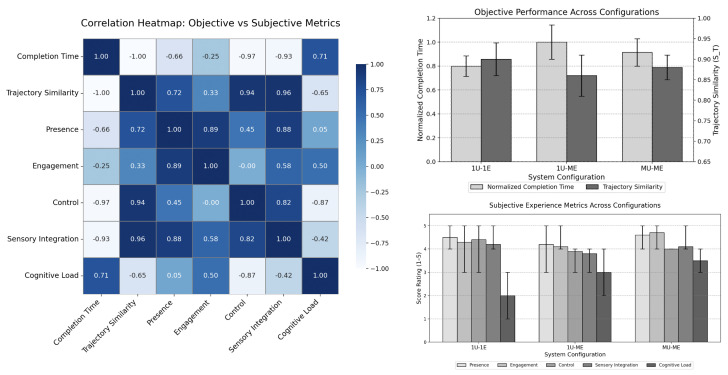
Performance trade-offs across system configurations. (**left**) Correlation heatmap showing the relationships between objective metrics (task completion time and trajectory similarity) and subjective metrics (presence, engagement, control, sensory integration, and cognitive load). (**top right**) Bar chart comparing normalized completion time ($T_{E}^{\mathrm{norm}}$) and trajectory similarity ($S_{T}$) across configurations (1U-1E, 1U-ME, and MU-ME), highlighting temporal-spatial efficiency differences. (**bottom right**) Aggregated user-reported Likert scores (1–5) with standard deviation bars for subjective experience dimensions across system configurations. The visualization emphasizes how minimal set-ups (1U-1E) favor control and efficiency, while multi-user set-ups (MU-ME) boost presence and engagement despite cognitive trade-offs.

## Data Availability

Data are contained within the article. The raw data supporting the conclusions of this article will be made available by the authors on request.

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
