# Peer review of "Immersive Teleoperation via Collaborative Device-Agnostic Interfaces for Smart Haptics: A Study on Operational Efficiency and Cognitive Overflow for Industrial Assistive Applications"

_sensors, 2025, doi:10.3390/s25133993_

Round 1
Reviewer 1 Report
Comments and Suggestions for Authors
This manuscript designs a device-agnostic teleoperation framework for immersive industrial applications. The framework supports various configurations and the authors evaluate user Quality of Experience (QoE) through both objective metrics and subjective metrics by comparing different haptic device setups. Results show significant differences in QoE across configurations, highlight the feasibility of the proposed device-agnostic teleoperation framework, and emphasize the interplay between task performance and user experience. The study focuses on immersive teleoperation with haptic feedback, which is an important and hot topic, but there are several key issues to be addressed.
- The device-agnostic layer described in section 3.2 is the core for the designed device-agnostic teleoperation framework. However, important details are missing in the description, such as the mentioned standardized communication protocols and scheduling algorithms. In addition, the description of the integration of haptic devices and trackers in section 3.3 is too concise and there are many repetitive modules in the Figures (Figure 4 ~ Figure 7) in section 3.3. Please reorganize these figures, for example, integrate them into one diagram, and provide more details about how to integrate these devices.
- The study recruited 40 participants. Whether the ethical review has been passed and whether the informed consent forms of the participants have been obtained? Please provide the details related to ethics.
- This work reports that the usage of body suit was optional on some sensitive cases with risk of cognitive overflow. Please provide details about which cases used the body suit and how to use it in the experiments.
- The work provides the statistical analysis results for temporal efficiency in sections 5.1,1. Has a similar statistical analysis been conducted for trajectory similarity?
- The experiments focus on objective and subjective QoE, which is important for users’experience and task execution. What about the performance of the designed teleoperation system, such as the latency, transmission speed, etc.
- This work just provides some pictures, which cannot display the real operation situation intuitively. Please provide some videos about the tasks used for the experiments.
- Other minor issues:
a. Line 240, 4x5 --> 4 × 5
b. Figure 10, please provide the meaning of the blue curves
c. Line 358 and 366, please number the equations and provide all the meanings of all symbols, such as “N”
d. Line 359 and 367, “where” should be placed on the left
Author Response
All comments from Reviewer 1 have been answered in the attached file.

Reviewer 2 Report
Comments and Suggestions for Authors
1. Why do the authors say current teleoperation systems often suffer from device-specific limitations and lack interoperability in Lines 50-51? This statement needs literature.
2. Most of the components in Figures 4- 7 are the same. It would be clearer if the differences between these figures were clarified.
3. Section 3.3 lists four types of devices that can be incorporated into the proposed architecture. However, it is more important to illustrate how to integrate different types of devices.
4. The methodology for evaluating QoE metrics is the first contribution. However, the metrics in Sections 4.3 and 4.2 are actually named with a new name that has the exact definition as those in other similar literature.
Author Response
All comments from Reviewer 2 have been answered in the attached file.

Reviewer 3 Report
Comments and Suggestions for Authors
Review of “Immersive Teleoperation via Collaborative Device-Agnostic Interfaces for Smart Haptics”
The authors address the issue of quality of experience (QoE) of teleoperation systems, in its objective and subjective aspects. The task of 40 subjects was to grasp simple objects (spheres, cylinders, etc.) with a teleoperated robot arm, and then to move them along a predefined 3D trajectory and place them into a bucket. The teleoperation system involved a data glove for haptic sensory feedback, and several controllers (grounded or mobile). The tasks were fulfilled in three system configurations: (1) single user operating one device at a time, (2) single user operating two or more devices simultaneously, or (3) two users two or more devices simultaneously.
The subject differed in expertise. Their performance was measured in terms of temporal efficiency, i. e. average task completion time, and accuracy. The latter was defined as cumulative deviation from the ideal trajectory and smoothness. In addition, subjective impressions of presence, engagement, etc. were taken.
It turned out that the grounded teleoperation device produced the shortest completion times. Robust haptic feedback was pretty much as good whereas the use of the joystick-like controller produced longer completion times, likely due to lag in the controller. The worst performance was produced with the glove and optical tracking device. And not surprisingly, the cooperative tasks took longer to complete. In terms of precision of the movement (trajectory similarity) the grounded device once again produced very good results, only exceeded by the combination of the sense glove with VIVE trackers.
I commend the authors on an outstanding manuscript. This is cutting edge research of great sophistication. The manuscript should definitely be published, one some Methods aspects are cleared up and once the subjective data are better presented.
Methods section: I believe that subjects had to produce ratings on scales ranging from 1 to 5. If this is correct, it should be made explicit when introducing the subjective dimensions.
Results section: The illustration of these rating averages in Figures 15, 16, and 17 are utterly confusing. For one, the ratings display a lot of variance such that box plots look ugly and confusing. Also, the total number of subjects is not really large enough to test group differences of expertise level. So I would suggest to have two figures here: One figure that displays bar graphs of each rating averaged across all observers per device. And one figure that shows averages per expertise group, averaged across tasks. The current figures confuse more than they help.
Discussion: I would like to see an evaluation of the similarity of performance with the four devices with respect to the concept of device-agnostic teleoperation. Do the authors take the differences to be small enough to support the concept of device-agnostic teleoperation, or do they take the concept to be a given and merely want to compare objective performance and subjective feelings of difficulty etc.?
In sum, this is a wonderful manuscript in which the illustrations of the subjective data need to be cleaned up.
Author Response
All comments from Reviewer 3 have been answered in the attached file.

Round 2
Reviewer 1 Report
Comments and Suggestions for Authors
The authors addressed all my concerns.
Reviewer 2 Report
Comments and Suggestions for Authors
All comments have been solved.